# CCPO: Constraint-Conditioned Policy Optimization for Versatile Safe Reinforcement Learning

**Yihang Yao**[*1]**, Zuxin Liu**[*1]**, Zhepeng Cen**[1]**,**
**Jiacheng Zhu**[1,3]**, Wenhao Yu**[2]**, Tingnan Zhang**[2]**, Ding Zhao**[1]
[1] Carnegie Mellon University, [2] Google DeepMind, [3] Massachusetts Institute of Technology
[*] Equal contribution, {yihangya, zuxinl}@andrew.cmu.edu

**Abstract:** Safe reinforcement learning (RL) focuses on training reward-maximizing agents subject to pre-defined safety constraints. Yet, learning versatile safe policies that can adapt to varying safety constraint requirements during deployment without retraining remains a largely unexplored and challenging area. In this work, we formulate the versatile safe RL problem and consider two primary requirements: training efficiency and zero-shot adaptation capability. To address them, we introduce the Conditioned Constrained Policy Optimization (CCPO) framework, consisting of two key modules: (1) Versatile Value Estimation (VVE) for approximating value functions under unseen threshold conditions, and (2) Conditioned Variational Inference (CVI) for encoding arbitrary constraint thresholds during policy optimization. Our extensive experiments demonstrate that CCPO outperforms the baselines in terms of safety and task performance while preserving zero-shot adaptation capabilities to different constraint thresholds data-efficiently. This makes our approach suitable for real-world dynamic applications.

**Keywords:** Safe Robot Learning, Multi-task Learning, Zero-shot Adaptation

## 1 Introduction

Safe reinforcement learning (RL) has emerged as a promising approach to address the challenges faced by robots operating in complex, real-world environments [1], such as autonomous driving [2], home service [3], and UAV locomotion [4]. Safe RL aims to learn a reward-maximizing policy within a constrained policy set [5, 6, 7]. By explicitly accounting for safety constraints during policy learning, agents can better reason about the trade-off between task performance and safety constraints, making them well-suited for safety-critic applications [8].

Despite the advances in safe RL, the development of a versatile policy that can adapt to varying safety constraint requirements during deployment without retraining remains a largely unexplored area. Investigating versatile safe RL is crucial due to the inherent trade-off between task reward and safety requirement [9, 10]: stricter constraints typically lead to more conservative behavior and lower task rewards. For example, an autonomous vehicle can adapt to different thresholds for driving on an empty highway and crowded urban area to maximize transportation efficiency. Consequently, learning a versatile policy allows agents to efficiently adapt to diverse constraint conditions, enhancing their applicability and effectiveness in real-world scenarios [11].

This paper studies the problem of training a versatile safe RL policy capable of adapting to tasks with different cost thresholds. The primary challenges are two-fold: **(1) Training efficiency.** A straightforward approach is to train multiple policies under different constraint thresholds, then the agent can switch between policies for different safety requirements. However, this method is sampling inefficient, making it unsuitable for most practical applications, as the agent may only collect data under a limited number of thresholds during training. **(2) Zero-shot adaptation capability.** Constrained optimization-based safe RL approaches rely on fixed thresholds during training [12], while recovery-based safe RL methods require a pre-defined backup policy to correct agent's unsafe

7th Conference on Robot Learning (CoRL 2023), Atlanta, USA.

behaviors [5, 6, 7]. Therefore, current safe RL training paradigms face challenges in adapting the learned policy to accommodate unseen safety thresholds. To tackle the challenges outlined above, we introduce the Conditioned Constrained Policy Optimization (CCPO) framework, a sampling-efficient algorithm for versatile safe reinforcement learning that achieves zero-shot generalization to unseen cost thresholds during deployment. Our key contributions are summarized as follows:

**1. We frame safe RL beyond pre-defined constraint thresholds as a versatile learning problem.** This perspective highlights the limitations of most existing constrained-optimization-based approaches and motivates the development of CCPO based on conditional variational inference. Importantly, CCPO can generalize to diverse unseen constraint thresholds without retraining the policy.

**2. We introduce two key techniques, Versatile Value Estimation (VVE) and Conditioned Variational Inference (CVI), for safe and versatile policy learning.** To the best of our knowledge, our method is the first successful online safe RL approach capable of achieving zero-shot adaptation for unseen thresholds while preserving safety. Our theoretical analysis further provides insights into our approach's data efficiency and safety guarantees.

**3. We conduct comprehensive evaluations of our method on various safe RL tasks.** The results demonstrate that CCPO outperforms baseline methods in terms of both safety and task performance for varying constraint conditions. The performance gap is notably larger in tasks with the high-dimensional state and action space, wherein all baseline methods fail to realize safe adaptation.

## 2 Problem Formulation

**Constrained Markov Decision Process:** CMDP $\mathcal{M}$ is defined by the tuple $(\mathcal{S}, \mathcal{A}, \mathcal{P}, r, c, \mu_0)$ [13], where $\mathcal{S}$ is the state space, $\mathcal{A}$ is the action space, $\mathcal{P}$ is the transition function, $r$ is the reward function, and $\mu_0$ is the initial state distribution. CMDP augments MDP with an additional element $c$ to characterize the cost of violating the constraint. Note that this work can be directly applied to the multiple-constraints setting, but we use a single constraint for ease of demonstration. Let $\pi$ denote the policy and $\tau = \{s_1, a_1, ...\}$ denote the trajectory. We use shorthand $\mathbf{f}_t = \mathbf{f}(s_t, a_t, s_{t+1}), \mathbf{f} \in \{r, c\}$ for simplicity. The value function is $V_{\mathbf{f}}^{\pi}(\mu_0) = \mathbb{E}_{\tau \sim \pi, s_0 \sim \mu_0}[\sum_{t=0}^{\infty} \gamma^t \mathbf{f}_t], \mathbf{f} \in \{r, c\}$, which is the expectation of discounted return under the policy $\pi$ and the initial state distribution $\mu_0$.

**Versatile safe RL problem:** Safe RL beyond a single pre-defined constraint threshold. Specifically, we consider a set of thresholds $\epsilon \in \mathcal{E}$ and a constraint-conditioned policy: $\pi(\cdot|\epsilon)$. We can then formulate the versatile safe RL problem as finding the optimal versatile policy $\pi^*(\cdot|\epsilon)$ that maximizes the reward within the corresponding threshold condition on a range of constraint thresholds $\epsilon \in \mathcal{E}$:

$$\pi^*(\cdot|\epsilon) = \arg\max_{\pi} V_r^{\pi}(\mu_0), \ s.t. \ V_c^{\pi}(\mu_0) \leq \epsilon, \ \forall \epsilon \in \mathcal{E}. \tag{1}$$

The training dataset $D = \bigcup_{i=1}^{N} D_i$ is collected through a limited set of thresholds $D_i \sim \pi(\cdot|\tilde{\epsilon}_i), \forall \epsilon_i \in \tilde{\mathcal{E}}$, with $\tilde{\mathcal{E}} \subset \mathcal{E}, |\tilde{\mathcal{E}}| = N$, where $N$ denotes the number of behavior policies with pre-specified constraint conditions during training. We also provide the related works in Appendix E.

## 3 Method

We identify two key challenges for versatile safe RL: (1) Q function estimation for unseen threshold conditions with limited behavior policy data and (2) encoding arbitrary safety constraint conditions in the versatile policy training. To address these challenges, we propose the Constraint-Conditioned Policy Optimization (CCPO) method.

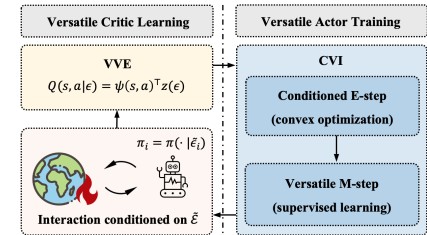

Figure 1: Proposed CCPO framework.

As illustrated in Figure. 1, CCPO operates in two modules. The first **versatile critic learning** involves the concurrent training of several behavior agents, each under their respective target constraint thresholds. The goal is to learn feature representations for both the state-action pair feature and the target thresholds, hence enabling generalization to unseen thresholds using Versatile Value Estimation (VVE). In the **versatile actor training**, we train the policy to be responsive to a range of unseen thresholds based on the well-trained value functions. Our

key insight is to adopt the variational inference safe RL framework [14, 15]. With this Conditional Variational Inference (CVI) step, the policy can achieve zero-shot adaptation to unseen thresholds without needing behavior agents to collect data under corresponding conditions. We introduce each module as follows.

## 3.1 Versatile value estimation

The estimation of Q-value functions becomes increasingly crucial when dealing with unseen thresholds that the behavior agents do not encounter. To tackle this problem, we propose the versatile critics learning module, which disentangles observations and target thresholds within a latent feature space. The assumption regarding the decomposition is as follows.

**Assumption 1** (Critics linear decomposition)**.** *The versatile Q functions $Q_{\mathbf{f}}^*$ with respect to the optimal versatile policy $\pi^*$ can be represented as:*

$$Q_{\mathbf{f}}^*(s, a|\epsilon) = \boldsymbol{\psi}_{\mathbf{f}}(s, a)^\top \boldsymbol{z}_{\mathbf{f}}^*(\epsilon), \quad \mathbf{f} \in \{r, c\}, \tag{2}$$

*where $||\boldsymbol{\psi}_{\mathbf{f}}(s, a)||_\infty \leq K_{\mathbf{f}}$ is the feature for the function of the state-action pair $(s, a)$, and $\boldsymbol{z}_{\mathbf{f}}^*$ is the optimal constraint-conditioned policy feature, which only depends on the policy condition $\epsilon$ for a specified task. The dimension of $\boldsymbol{\psi}_{\mathbf{f}}(s, a)$ and $\boldsymbol{z}_{\mathbf{f}}^*(\epsilon)$ are both $M$.*

Here we decompose the Q function into the product of $\boldsymbol{\psi}(s, a)$ and $\boldsymbol{z}(\epsilon)$. In practice, we utilize a Neural Network to parameterize $\boldsymbol{\psi}(s, a)$ and a much smaller model such as polynomial regression to parameterize $\boldsymbol{z}(\epsilon)$. Thus we can get a bounded estimation error for Q functions $\hat{Q}_{\mathbf{f}}$ under unseen threshold conditions. The theoretical analysis of bounded estimation, empirical verification of Q function learning, and the model structure can be found in Appendix B.1, A.5, D.5, respectively.

## 3.2 Conditioned variational inference

Given well-trained versatile Q functions in versatile critics learning, we aim to encode arbitrary threshold constraints during policy learning in the versatile actor training. We utilize the *safe RL as inference* framework to achieve this goal. The **key strength** of using this framework lies in its ability to encode arbitrary threshold conditions during policy learning, as shown in (3), a feat that is challenging for other methods, such as those based on primal-dual algorithms [14]. Following the *EM-style* policy optimization for RL as inference, our Constrained variational inference (CVI) realizes the versatile policy update via Constraint-conditioned E-step and Versatile M-step as follows.

**Constraint-Conditioned E-step:** The conditioned E-step aims to find the optimal variational distribution $q(a|s, \epsilon_i)$ that maximizes the reward return while satisfying the safety condition defined by $\epsilon_i$ for the state $s$ and resampled action $a$. At the $j$-th iteration, denote the policy parameter as $\theta_j$, we can write the policy update objective w.r.t $q$ as a constrained optimization problem:

$$\max_{q(a|s, \epsilon_i)} \mathbb{E}_{\rho_q} \left[ \int q(a|s, \epsilon_i) \hat{Q}_r^{\pi_{\theta_j}}(s, a|\epsilon_i) da \right] \quad \text{s.t.} \quad \mathbb{E}_{\rho_q} \left[ \int q(a|s, \epsilon_i) \hat{Q}_c^{\pi_{\theta_j}}(s, a|\epsilon_i) da \right] \leq \epsilon_i, \tag{3}$$

where $\hat{Q}_{\mathbf{f}}(\cdot|\epsilon_i)$ is the versatile Q functions as introduced in section 3.1, the inequality constraint represents the safety constraint defined by $\epsilon_i$. The closed-form solution $q_i^*(a|s, \epsilon_i)$, which means the optimal action distribution, can be found in Appendix C.3.

**Versatile M-step:** After the constraint-conditioned E-step, we obtain a set of optimal feasible variational distribution $q_i^* = q_i^*(\cdot|s, \epsilon_i)$ for each constraint threshold $\epsilon_i$. In the versatile M-step, we aim to improve the policy (see objective details in the Appendix A.4) w.r.t the policy parameter $\theta$ for $\epsilon_i \in \mathcal{E}$, which is a supervised-learning problem with KL-divergence constraints [16, 17]:

$$\max_{\theta} \mathbb{E}_{\rho_q} \left[ \sum_{i=1}^{|\mathcal{E}|} \mathbb{E}_{q_i^*} \left[ \log \pi_\theta(a|s, \epsilon_i) \right] / |\mathcal{E}| \right] \quad \text{s.t.} \quad \mathbb{E}_{\rho_q} \left[ D_{\mathrm{KL}}(\pi_{\theta_j}(a|s, \epsilon_i) \| \pi_\theta(a|s, \epsilon_i)) \right] \leq \gamma \, \forall i, \tag{4}$$

where $\mathcal{E}$ is the set for all the sampled versatile policy conditions $\epsilon_i$ in fine-tuning stage of training. The constraint in (4) is a regularizer to stabilize the policy update.

## 4 Experiments

**Baselines:** We divide the baseline methods into two categories and name them as: **Constraint-conditioned baselines.** (V-SAC, and V-DDPG), and **policy linear combination** (C-PPO, and C-TRPO)

**Task.** The simulation environments are from a publicly available benchmark [18]. We name the tasks as `Ball-Circle` (BC), `Car-Circle` (CC), `Drone-Circle` (DC), `Drone-Run` (DR), and `Ant-Run` (AR) [10, 11, 12]. More details about baselines and tasks can be found in Appendix D.1.

**Metrics:** We compare the methods in terms of episodic reward (the higher, the better) and episodic constraint violation cost (the lower, the better) on each evaluated threshold condition. We take the average of the episodic reward and episodic cost as the main comparison metrics. For the reward, the higher the better. For the cost, it should be less than the target threshold condition.

The evaluation results are shown in Figure. 2. The models are trained on $\tilde{\mathcal{E}} = \{20, 40, 60\}$ and evaluated on $\mathcal{E}_g = \{10, 15, ..., 70\}$, and we report the averaged reward and constraint violation values on $\mathcal{E}_g$. Each value is reported for 50 episodes and 5 seeds.

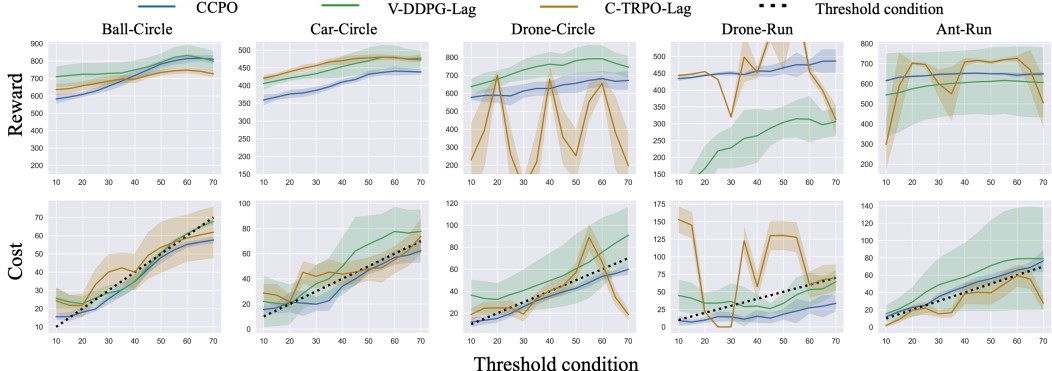

Figure 2: Results of zero-shot adaption to different cost returns. Each column is a task. The x-axis is the threshold condition. The first row shows the evaluated reward and the second row shows the evaluated cost under different target costs. The solid line is the mean value, and the light shade represents the area within one standard deviation. The versatile agents are trained on $\tilde{\mathcal{E}} = \{20, 40, 60\}$, and evaluated on $\mathcal{E}_g = \{10, 15, ..., 70\}$.

For baseline methods, we can observe that in the `Drone-Circle`, `Drone-Run`, and `Ant-Run` tasks characterized by highly-nonlinear robot dynamics, the **constraint-conditioned baseline** method (`V-DDPG`) exhibits the poor ability to encode threshold into the policy generation, thus leading to high cost violation values. This limitation arises due to the inadequacy of utilizing only a limited number of behavior policies for versatile policy training in tasks with high-dimensional observation and action spaces. for the **policy linear combination** baseline (`C-TRPO`), it has a significant reward drop at unseen thresholds. It indicates the concepts from the control theory that the safety-critical control component is proportional to the conservativeness level and can not be directly used in versatile safe RL with high-dimensional settings.

For the proposed CCPO method, we can clearly see that **CCPO learns a versatile safe RL policy that can generalize well to unseen thresholds data-efficiently** with low cost-violations and high rewards. We also provide detailed full quantitative experiment results, more baseline comparison, results for the ablation study, and the different choices of behavior policies in Appendix D.

## 5 Conclusion

In this study, we pioneered the concept of versatile safe reinforcement learning (RL), presenting the Conditioned Constrained Policy Optimization (CCPO) algorithm. This approach adapts efficiently to different and unseen cost thresholds, offering a promising solution to safe RL beyond pre-defined constraint thresholds. With its core components, Versatile Value Estimation (VVE) and Conditioned Variational Inference (CVI), CCPO facilitates zero-shot generalization for constraint thresholds. Our theoretical analysis further offers insights into the constraint violation bounds for unseen thresholds and the sampling efficiency of the employed behavior policies. The extensive experimental results reconfirm that CCPO effectively adapts to unseen threshold conditions and is much safer and more data-efficient than baseline methods.

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

# Contents

# A Algorithm details

## A.1 Details for Versatile value estimation

Note that Assumption 1 is reasonable and widely accepted in the RL transfer learning literature with successor features [19], as it is reasonable to find a high-dimensional feature space to decompose the Q functions into the product of feature functions $\psi_{\mathbf{f}}(s, a)$ and the latent vectors $z_{\mathbf{f}}(\epsilon)$ [20]. As shown in Theorem 1, by learning $\psi_{\mathbf{f}}(s, a)$ and $z_{\mathbf{f}}(\epsilon)$ jointly and adding norm constraints on the feature function $||\psi_{\mathbf{f}}(s, a)||_{\infty} \leq K_{\mathbf{f}}$, we can efficiently encode the threshold information $\epsilon$ into the Q functions and achieve accurate estimations for unseen thresholds. This is the basis for our method's data-efficient training. To further facilitate theoretical analysis, we assume that the optimal constraint-conditioned policy feature $z_{\mathbf{f}}^*(\epsilon)$ can be approximated by polynomial functions:

**Assumption 2** (Polynomial feature space)**.** *The optimal constraint-conditioned policy feature $z_{\mathbf{f}}^*$ can be approximated by $z_{\mathbf{f}}^*(\epsilon) = Poly(\epsilon, p) + e$, meaning each element of $z_{\mathbf{f}}^*$ is a $p$-degree polynomial of $\epsilon$, and $e$ is the remainder. Each component for $e$ follows $e_j \sim \mathcal{N}(0, \sigma_j^2), j = i, ..., M$, and denote $\sigma = \max_j \sigma_j$.*

Note that the degree $p$ corresponds to the $z(\epsilon)$ model representation capability. Based on the above assumptions, we can derive the Q function estimation error bound as follows.

**Theorem 1** (Bounded estimation error)**.** *Denote $\epsilon_L$ and $\epsilon_H$ are the lower and upper bound of the target threshold interval for $\mathcal{E}$. Suppose the threshold conditions $\{\tilde{\epsilon}_i\}_{i=1,2,...,N}$ for behavior policies are selected to divide the interval $[\epsilon_L, \epsilon_H]$ evenly, then with confidence level $1 - \alpha$, the estimation error of versatile Q functions conditioned on arbitrary $\epsilon \in [\epsilon_L, \epsilon_H]$ can be bounded by:*

$$||\hat{Q}_{\mathbf{f}}(s, a|\epsilon) - Q_{\mathbf{f}}^*(s, a|\epsilon)|| \leq \frac{z_{\alpha/2}B(p)}{N^{\beta(p)}}\sqrt{\sigma^2 K_{\mathbf{f}}^2 M}, \tag{5}$$

where $B(p)$ and $\beta(p)$ are both functions of the polynomial degree $p$, and $z_{\alpha/2}$ is the upper alpha quantile for the standard Gaussian distribution. The proof and detailed discussion of Theorem 1 and functions $B(p), \beta(p)$ are shown in Appendix C.1. It is worth noting that we normalize the threshold conditions $\epsilon \in [\epsilon_L, \epsilon_H]$ to the interval $[0, 1]$ by $\epsilon = (\epsilon - \epsilon_L)/\epsilon_H$ for numerical stability. Theorem 1 establishes that by decomposing the Q function into the product of $\psi(s, a)$ and $z(\epsilon)$ and jointly learning these components, we can guarantee a bounded estimation error for Q functions under unseen threshold conditions. Furthermore, we can derive the bounded cost violation for unseen thresholds and $\epsilon$-sample complexity analysis based on the theorem, both of which are discussed in proposition 1 and remark 1. We also provide empirical verification of Q function estimation in Appendix D.5.

## A.2 Derivation of ELBO

We utilize the *safe RL as inference* framework to achieve this goal, as it decomposes safe RL to a convex optimization followed by supervised learning, both stages readily accommodating varying target thresholds. In contrast to the classical view of safe RL aiming to find the most-rewarding actions while satisfying the constraints, the probabilistic inference perspective finds the feasible (constraint-satisfying) actions most likely to have been taken given future success in maximizing task rewards [14].

Following the RL as inference literature [21], we consider an infinite discounted reward formulation. In the condition that the constraint threshold is $\epsilon_i$, we denote $O = O(s, a)$ as the optimality variable of a state-action pair $(s, a)$, which indicates the reward-maximizing (optimal) event by choosing an action $a$ at a state $s$. Then for a given trajectory $\tau$, the likelihood of being optimal is proportional to the exponential of the discounted cumulative reward: $p(O = 1|\tau) \propto \exp(\sum_t \gamma^t r_t/\alpha)$, where $\alpha$ is a temperature parameter. Since the probability of getting a trajectory $\tau$ under the conditioned policy $\pi(\cdot|\epsilon_i)$ can be expressed as $p_{\pi(\cdot|\epsilon_i)}(\tau) = p(s_0)\prod_{t \geq 0} p(s_{t+1}|s_t, a_t)\pi(a_t|s_t, \epsilon_i)$, the lower bound for

the log-likelihood of optimality given the conditioned policy $\pi(\cdot|\epsilon_i)$ is:

$$
\begin{aligned}
\log p_{\pi(\cdot|\epsilon_i)}(O=1) &= \log \mathbb{E}_{\tau \sim q(\cdot|\epsilon_i)} \frac{p(O=1|\tau)p_\pi(\tau|\epsilon_i)}{q(\tau|\epsilon_i)} \\
&\geq \mathbb{E}_{\tau \sim q(\cdot|\epsilon_i)} \log \frac{p(O=1|\tau)p_{\pi(\cdot|\epsilon_i)}(\tau)}{q(\tau|\epsilon_i)} \\
&\propto \mathbb{E}_{\tau \sim q(\cdot|\epsilon_i)}[\sum_{t=0}^{\infty}\gamma^t r_t] - \alpha D_{\mathrm{KL}}(q(\tau|\epsilon_i)\|p_{\pi(\cdot|\epsilon_i)}(\tau)) := \mathcal{J}(q,\pi|\epsilon_i),
\end{aligned}
\tag{6}
$$

where the inequality follows Jensen's inequality, and $q(\tau|\epsilon_i)$ is an auxiliary trajectory-wise variational distribution conditioned on $\epsilon_i$. $\mathcal{J}(q,\pi|\epsilon_i)$ in equation (6) is the evidence lower bound (ELBO) to reach the reward optimality under condition $\epsilon_i$. Since $q(\tau|\epsilon_i) = p(s_0)\prod_{t\geq 0}p(s_{t+1}|s_t,a_t)q(a_t|s_t,\epsilon_i)$, we have the following ELBO over the state and constraint conditioned action distribution $q(a|s,\epsilon_i)$:

$$
\mathcal{J}(q,\theta|\epsilon_i) = \mathbb{E}_{\rho_q(\cdot|\epsilon_i)}\Big[\sum_{t=0}^{\infty}\gamma^t r_t - \alpha D_{\mathrm{KL}}(q(\cdot|\epsilon_i)\|\pi_\theta(\cdot|\epsilon_i))\Big] + \log p(\theta)
\tag{7}
$$

where $\rho_q(s|\epsilon_i)$ is the stationary state distribution induced by $q(\cdot|s,\epsilon_i)$ and $\rho_0$, $\theta$ refers to the parameters for policy $\pi$, and $p(\theta)$ is a prior distribution over the parameters. Note we overload $q$ by using it both in $q(a|s,\epsilon_i)$ and $q(\tau|\epsilon_i)$.

### A.3 Constraint-conditioned E-step details

The conditioned E-step aims to find the optimal variational distribution $q(\cdot|\epsilon_i) \in \Pi_{\mathcal{Q}}^{\epsilon_i}$ that maximizes the reward return while satisfying the safety condition defined by $\epsilon_i$. At the $j$-$th$ iteration, We can write the ELBO objective w.r.t $q$ as a constrained optimization problem (see Appendix C for proofs):

$$
\begin{aligned}
\max_{q(a|s,\epsilon_i)} &\ \mathbb{E}_{\rho_q}\left[\int q(a|s,\epsilon_i)\hat{Q}_r^{\pi_{\theta_j}}(s,a|\epsilon_i)\,da\right] \\
\text{s.t. } &\ \mathbb{E}_{\rho_q}\left[\int q(a|s,\epsilon_i)\hat{Q}_c^{\pi_{\theta_j}}(s,a|\epsilon_i)\,da\right] \leq \epsilon_i, \\
&\ \mathbb{E}_{\rho_q}\left[D_{\mathrm{KL}}\left(q(a|s,\epsilon_i)\|\pi_{\theta_j}(\cdot|\epsilon_i)\right)\right] \leq \kappa;
\end{aligned}
\tag{8}
$$

where $\hat{Q}_\mathbf{f}(\cdot|\epsilon_i)$ is the versatile Q functions as introduced in section 3.1, the first inequality constraint represents the safety constraint and the last term in the constraint is the trust region with the old policy defined by KL distance $\kappa$. Inspired by [16], we use the solution of the optimal variational distribution $q_i^* = q_i^*(a|s,\epsilon_i)$ for arbitrary safety constraint $\epsilon_i$, which has the closed form:

$$
q_i^* = \frac{\pi_{\theta_j}(\cdot|\epsilon_i)}{Z(s,\epsilon_i)}\exp\left(\frac{\hat{Q}_r^{\pi_{\theta_j}}(\cdot|\epsilon_i) - \lambda_i^*\hat{Q}_c^{\pi_{\theta_j}}(\cdot|\epsilon_i)}{\eta_i^*}\right),
\tag{9}
$$

where $Z(s,\epsilon_i)$ is a normalizer to make sure $q_i^*$ is a valid distribution, and the dual variables $\eta_i^*$ and $\lambda_i^*$ are the solutions of the following convex optimization problem (see Appendix C for details):

$$
\min_{\lambda_i,\eta_i \geq 0}\ g(\eta_i,\lambda_i) = \lambda_i\epsilon_i + \eta_i\kappa\mathbb{E}_{\rho_q}\left[\log\mathbb{E}_{\pi(\cdot|\epsilon_i)}\left[\exp\left(\frac{\hat{Q}_r(\cdot|\epsilon_i) - \lambda_i\hat{Q}_c(\cdot|\epsilon_i)}{\eta_i}\right)\right]\right].
\tag{10}
$$

Then we can encode arbitrary safety constraints by calculating the optimal distribution $q_i^*(a|s,\epsilon_i)$ regarding the corresponding condition $\epsilon_i$ efficiently with (9). The term $q_i^*(a|s,\epsilon_i)$ means when conditioned on $\epsilon_i$, the probability of taking $a$ at $s$ for the optimal feasible policy.

### A.4 Versatile M-step details

After the constraint-conditioned E-step, we obtain a set of optimal feasible variational distribution $q_i^* = q_i^*(\cdot|s,\epsilon_i)$ for each constraint threshold $\epsilon_i$. In the versatile M-step, we aim to improve the ELBO (6) w.r.t the policy parameter $\theta$ for $\epsilon_i \in \mathcal{E}$.

$$
\mathcal{J}(\theta|\epsilon_i) = \mathbb{E}_{\rho_q}\Big[\alpha\mathbb{E}_{q_i^*}\big[\log\pi_\theta(a|s,\epsilon_i)\big]\Big] + \log(p|\epsilon_i)
\tag{11}
$$

Using a Gaussian prior for each threshold-conditioned policy, this problem can be further converted to the following supervised-learning problem with KL-divergence constraints [16, 17]:

$$\max_\theta \mathbb{E}_{\rho_q} \Big[ \sum_{i=1}^{|\mathcal{E}|} \mathbb{E}_{q_i^*} \big[ \log \pi_\theta(a|s, \epsilon_i) \big] / |\mathcal{E}| \Big] \ s.t. \ \mathbb{E}_{\rho_q} \big[ D_{\mathrm{KL}}(\pi_{\theta_j}(a|s, \epsilon_i) \| \pi_\theta(a|s, \epsilon_i)) \big] \leq \gamma \ \forall i, \quad (12)$$

where $\mathcal{E}$ is the set for all the sampled versatile policy conditions $\{\epsilon_i\}$ in fine-tuning stage of training. The constraint in (4) is a regularizer to stabilize the policy update.

## A.5 CCPO implementation details

Due to the page limit, we omit the implementation details of CCPO in the main content. We will present the full algorithm and some implementation tricks in this section. Without otherwise statements, the critics' and policies' parametrization is assumed to be neural networks (NNs), while we believe other parametrization forms should also work in practice.

**Critics update**. Denote $\phi_{\psi_{\mathbf{f}}}, \phi_{z_{\mathbf{f}}}$ as the parameters for $\psi_{\mathbf{f}}(s, a)$ and $z_{\mathbf{f}}(\epsilon)$ in the critic $Q_{\mathbf{f}}(s, a|\epsilon)$. Similar to many other off-policy algorithms [22], we use a target network for each critic and the polyak smoothing trick to stabilize the training. Other off-policy critic's training methods, such as Re-trace, could also be easily incorporated with the CCPO training framework. Denote $\phi'_{\psi_r}, \phi'_{z_r}$ as the parameters for the **target** reward critic $Q'_r$, and $\phi'_{\psi_c}, \phi'_{z_c}$ as the parameters for the **target** cost critic $Q'_c$. Define $\mathcal{D} = \cup_{\tilde\epsilon_i \in \tilde{\mathcal{E}}} \mathcal{D}_i$ as the replay buffer and $(s, a, s', r, c, \tilde\epsilon_i)$ as the state, action, next state, reward, cost, and behavior policy condition, respectively. The critics are updated by minimizing the following mean-squared Bellman error (MSBE):

$$L(\phi_r) = \sum_{\mathcal{D}_i} \mathbb{E}_{(s,a,s',r,c) \sim \mathcal{D}_i} \Big[ \big( Q_r(s, a|\epsilon_i) - (r + \gamma \mathbb{E}_{a' \sim \pi(\cdot|\epsilon_i)}[Q'_r(s', a'|\epsilon_i)]) \big)^2 \Big] \quad (13)$$

$$L(\phi_c) = \sum_{\mathcal{D}_i} \mathbb{E}_{(s,a,s',r,c) \sim \mathcal{D}_i} \Big[ \big( Q_c(s, a|\epsilon_i) - (c + \gamma \mathbb{E}_{a' \sim \pi(\cdot|\epsilon_i)}[Q'_c(s', a'|\epsilon_i)]) \big)^2 \Big]. \quad (14)$$

Denote $\alpha_c$ as the critics' learning rate, we have the following updating equations:

$$\phi_{\psi_{\mathbf{f}}} \leftarrow \phi_{\psi_{\mathbf{f}}} - \alpha_c \nabla_{\phi_r} L(\phi_r), \quad \phi_{z_{\mathbf{f}}} \leftarrow \phi_{z_{\mathbf{f}}} - \alpha_c \nabla_{\phi_c} L(\phi_c). \quad (15)$$

We use the polyak averaging trick to update the critics with a weight parameter $\rho \in (0, 1)$:

$$\phi'_{\psi_{\mathbf{f}}} = \rho \phi'_{\psi_{\mathbf{f}}} + (1 - \rho) \phi_{\psi_{\mathbf{f}}} \quad \phi'_{z_{\mathbf{f}}} = \rho \phi'_{z_{\mathbf{f}}} + (1 - \rho) \phi_{z_{\mathbf{f}}}. \quad (16)$$

**Full Algorithm**. Note that for off-policy methods, we need to convert the episodic-wise constraint violation threshold to a state-wise threshold for the $Q_c$ functions. Denote $T$ as the episode length, the target cost limit for one episode is $\epsilon_i^T$. Denote the discounting factor as $\gamma$. Then, if we assume that at each time step we have an equal probability to violate the constraint, the target constraint value $\epsilon_i$ for safety critic $Q_c^{\pi_{\theta_j}}(\cdot|\epsilon_i)$ could be approximated by:

$$\epsilon_i = \epsilon_i^T \times \frac{1 - \gamma^T}{T(1 - \gamma)}$$

The converted threshold $\epsilon_i$ will be used to compute the Lagrangian multipliers for the baselines, and also be used as one of the constraint thresholds in the constraint-conditioned E-step of our method:

$$\int \pi(a|s, \epsilon_i) \hat{Q}_c^{\pi_{\theta_j}}(s, a|\epsilon_i) \leq \epsilon_i, \quad \forall s, a$$

More details can be found in the code.

**Model structure.** The versatile critics model $Q_{\mathbf{f}}(s, a|\epsilon_i) = \psi_{\mathbf{f}}(s, a)^\top z_{\mathbf{f}}(\epsilon_i)$ is shown in Figure. 3. We set the feature dimension $M = 32$, and use an MLP with size [256, 256] to map from the state-action pair to the feature $\psi_{\mathbf{f}}$. Also, we use an MLP with size [32, 32] to map from $\epsilon$ to $z_{\mathbf{f}}$. For the versatile actor, we provide two options for the model structure. The first one is to direct concat the threshold $\epsilon_i$ into the state: $\bar{s} = [s, \epsilon_i]$ (Con) as shown in Figure. 4. The second one is to use the Multiplicative Interaction (MI) structure inspired by previous works [23, 24, 25, 26]. To give a fair comparison, we use the Con net for both our methods and the baseline. The user may turn on the MI option in the code to get a higher performance of our proposed CCPO method.

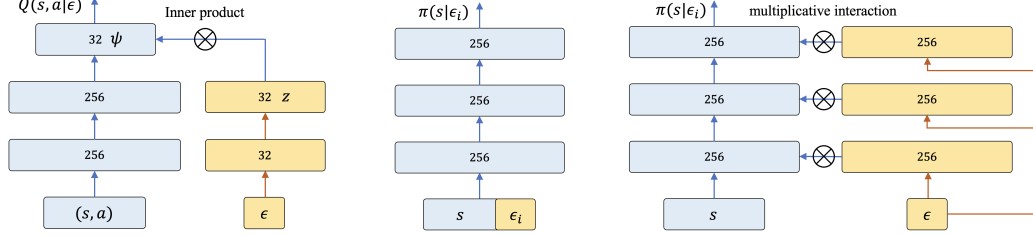

Figure 3: Versatile critics using Linear decomposition.

Figure 4: Versatile actor using state augmentation.

Figure 5: Versatile actor using multiplicative interaction.

# B Supplementary theoretical analysis

## B.1 Bounded safety violation

**Proposition 1** (Bounded safety violation). *With the threshold conditions $\tilde{\epsilon}_i \in \tilde{\mathcal{E}}$ for behavior policies selected to divide the target condition interval $[\epsilon_L, \epsilon_H]$ evenly, and with confidence level $1 - \alpha$, the cost violation of versatile policy under arbitrary threshold condition $\epsilon \in [\epsilon_L, \epsilon_H]$ is bounded as:*

$$V_c^{\pi(\mu_0|\epsilon)} - \epsilon \leq \frac{z_{\alpha/2} B(p)}{N^{\beta(p)}} \sqrt{\sigma^2 K_c^2 M}, \qquad (17)$$

The proof is shown in Appendix C.1. Proposition 1 ensures that the cost violation of the versatile safe RL agent on unseen thresholds can be bounded if the selected behavior policy conditions divide the interval $[\epsilon_L, \epsilon_H]$ evenly. We can observe that the bound (17) is proportional to $\sqrt{K_c^2 M}$. Since larger $K_c$ and $M$ correspond to a wider range of threshold conditions, this safety violation bound is related to the interval range. Also, we provide the complexity analysis for the $\epsilon$-sample, i.e., the estimation error corresponding to the number of behavior policies $N$ as shown in remark 1.

## B.2 $\epsilon$-sample complexity analysis

**Remark 1** ($\epsilon$-sample complexity analysis). *The estimation error for Q functions and safety violation bound decreases as the number of behavior policies $N$ increases. The decreasing rate is proportional to $\frac{1}{N^{\beta(p)}}$, where the exponent of $N$ is related to $p$, which is the representation capabilities of the model to represent the constraint-conditioned policy feature $z(\epsilon)$. When $p$ increases, $\beta$ also increases as shown in the Appendix C.1, which means when the model capability is high, the proposed method significantly becomes more data-efficient.*

The functions $\beta(p), B(p)$ are shown in Appendix C.1 and the proof is shown in Appendix C.2. Since CCPO is under the "RL as inference" framework, we also enjoy many benefits as revealed in previous works, such as the optimality guarantees and training robustness [14].

## B.3 Optimality

The optimality guarantee is mainly derived from the EM-style safe RL training framework, and more details could be found in Appendix A.5 in [14]. For self-contained, we briefly introduce it as follows.

**Assumption 3** (Slater's condition). *There exists a feasible distribution $\bar{q} \in \Pi_Q^{\epsilon_i}$ within the trust region of the old policy $\pi_{\theta_j}(\cdot|\epsilon_i)$: $D_{\mathrm{KL}}(\bar{q}\|\pi_{\theta_j}(\cdot|\epsilon_i)) < \kappa, \forall \epsilon_i \in \mathcal{E}$.*

**Assumption 4** (Two-step Slater's condition). *The Slater's condition holds for both $\pi_{\theta_{j-1}}$ and $\pi_{\theta_j}$.*

The above assumptions indicate a well-optimized policy in the constraint-conditioned M-step. In addition, they ensure the variational distributions $q_{j-1}(\cdot|\epsilon_i)$ and $q_j(\cdot|\epsilon_i)$ are feasible. Note that an infeasible variational distribution $q_{j-1}(\cdot|\epsilon_i)$ may lead to arbitrarily high reward return. With this assumption, we separately prove the ELBO improvement for constraint-conditioned E-step and versatile M-step.

**Proposition 2** (Optimality guarantee). *Suppose the optimal distribution at $j-1$-th and $j$-th update $\pi_{\theta_{j-1}}, \pi_{\theta_j}$ both satisfy the Slater's condition, then the ELBO in Eq. (7) is guaranteed to be non-decreasing: $\mathcal{J}(q_j, \theta_{j+1}|\epsilon_i) \geq \mathcal{J}(q_{j-1}, \theta_j|\epsilon_i)$.*

*Proof.* **Constraint-conditioned E-step**: By the definition of constrained-condition E-step, we improve the ELBO w.r.t $q(\cdot|\epsilon_i)$. Since $q_{j-1}(\cdot|\epsilon_i)$ is feasible (reward return is bounded) and $\pi_{\theta_j}(\cdot|\epsilon_i)$ satisfies Slater's condition, we can prove the closed-form solution of the constrain-conditioned E-step

update will increase ELBO:

$$
\begin{cases}
q_j(\cdot|\epsilon_i) = \underset{q\in\Pi_{\mathcal{Q}}^{\epsilon_i}}{\arg\max}\, \mathbb{E}_{\rho_q}\left[\mathbb{E}_{a\sim q(\cdot|s,\epsilon_i)}\left[\hat{Q}_r^{\pi_{\theta_j}}(s,a|\epsilon_i)\right] - \alpha D_{\mathrm{KL}}[q(\cdot|s,\epsilon_i)\|\pi_{\theta_i}(\cdot|s,\epsilon_i)]\right] \\
\qquad = \underset{q\in\Pi_{\mathcal{Q}}^{\epsilon_i}}{\arg\max}\, \mathbb{E}_{\tau\sim q}\left[\sum_{t=0}^{\infty}\left(\gamma^t r_t - \alpha D_{\mathrm{KL}}(q(\cdot|s_t,\epsilon_i)\|\pi_{\theta_j}(\cdot|s_t,\epsilon_i))\right)\right] \\
\qquad = \underset{q\in\Pi_{\mathcal{Q}}^{\epsilon_i}}{\arg\max}\, \mathcal{J}(q,\theta_j|\epsilon_i) \\
q_{i-1}(\cdot|\epsilon_i) \in \Pi_{\mathcal{Q}}^{\epsilon_i}
\end{cases}
$$
$$\Rightarrow \mathcal{J}(q_j,\theta_j|\epsilon_i) \geq \mathcal{J}(q_{j-1},\theta_j|\epsilon_i).$$

Therefore, as long as assumption 4 holds, the ELBO will increase monotonically in terms of $q$.

**Versatile M-step**: Assuming the ELBO conditioned on different constraints $\epsilon_i \in \mathcal{E}$ are independent of $\theta$. By definition in Eq. (4), when conditioned on $\epsilon_i$, we update $\theta$ by

$$
\begin{aligned}
\theta_{j+1} &= \underset{\theta}{\arg\max}\, \mathbb{E}_{\rho_{q_j}}\left[\alpha\mathbb{E}_{a\sim q_j(\cdot|s,\epsilon_i)}\left[\log\pi_\theta(a|s,\epsilon_i)\right]\right] + \log p(\theta) \\
&= \underset{\theta}{\arg\max}\, \mathbb{E}_{\rho_{q_j}}\left[-\alpha D_{\mathrm{KL}}[q_i(\cdot|s_t,\epsilon_i)\|\pi_\theta(\cdot|s,\epsilon_i)]\right] + \log p(\theta) \\
&= \underset{\theta}{\arg\max}\, \mathbb{E}_{\rho_{q_j}}\left[\mathbb{E}_{a\sim q_j(\cdot|s,\epsilon_j)}\left[\hat{Q}_r^{q_j}(s,a|\epsilon_i)\right] - \alpha D_{\mathrm{KL}}[q_j(\cdot|s_t,\epsilon_i)\|\pi_\theta(\cdot|s,\epsilon_i)]\right] + \log p(\theta) \\
&= \underset{\theta}{\arg\max}\, \mathcal{J}(q_j,\pi_\theta|\epsilon_i).
\end{aligned}
$$

Therefore, we have: $\mathcal{J}(q_j,\pi_{\theta_{j+1}}|\epsilon_i) \geq \mathcal{J}(q_j,\pi_{\theta_j}|\epsilon_i)$. Combining all the above together, we have

$$\mathcal{J}(q_i,\pi_{\theta_{j+1}}|\epsilon_i) \geq \mathcal{J}(q_i,\pi_{\theta_j}|\epsilon_i) \geq \mathcal{J}(q_{i-1},\pi_{\theta_j}|\epsilon_i).$$

$\square$

### B.4 Training robustness

The training robustness is mainly derived from the EM-style safe RL training framework, and more details could be found in Appendix A.6 in [14]. For self-contained, we briefly introduce it as follows.

**Proposition 3** (Training robustness). *Suppose $\pi_{\theta_j}(\cdot|\epsilon_i) \in \Pi_{\mathcal{Q}}^{\epsilon_i}$. $\pi_{\theta_{j+1}}$ and $\pi_{\theta_j}$ are related by the M-step. If $\epsilon_i < \gamma$, where $\epsilon_i, \gamma$ are the KL threshold in constraint-conditioned E-step and versatile M-step respectively, then the variational distribution $q_{j+1}^*$ in the next iteration is guaranteed to be feasible and optimal.*

*Proof.* Since $\epsilon < \epsilon_2$, the KL divergence between $\pi_{\theta_{i+1}}$ and $\pi_{\theta_i}$ $D_{\mathrm{KL}}(\pi_{\theta_i}\|\pi_{\theta_{i+1}}) \leq \epsilon < \epsilon_2$. Thus, the Slater condition 3 holds for $\pi_{\theta_{i+1}}$ as long as $\pi_{\theta_i}$ is feasible, because at least one feasible solution $\pi_{\theta_i}$ within the trust region exists. In Appendix. C.3, we know that $q_{i+1}^*(\cdot|\epsilon_i)$ in the constraint-conditioned E-step is guaranteed to be feasible and optimal. $\square$

## C   Proofs and discussions

### C.1   Proof of Theorem 1

*Proof.* In the proof, we omit the subscript $\mathbf{f} \in \{r, c\}$ for notation simplicity. The results hold for both cost and reward Q functions. Denote $\boldsymbol{x_j} = [1, \tilde{\epsilon}_j, ..., \tilde{\epsilon}_j^p]$, $\boldsymbol{X} = [\boldsymbol{x_1}^T, ..., \boldsymbol{x_N}^T] \in \mathbb{R}^{(p+1) \times N}$. For each component $z_i$ of $\boldsymbol{z}$, it can be written as:

$$z_i^* = \boldsymbol{\beta_i}^T \boldsymbol{x} + e_i; \quad i = 1, ..., M \tag{18}$$

where $e_i$ is the remainder, and we regard it following the Gaussian distribution: $e_i \sim \mathcal{N}(0, \sigma_i^2)$, and $\sigma_i \leq \sigma$. We divide the proof of the bounded estimation error into the following parts:

**(1) Estimation error bound depending on the threshold $\epsilon$ for generalization:** The objective of the Poly-regression is to find the optimal parameter $\beta_i$ such that it minimizes the Mean Squared Error:

$$\hat{\boldsymbol{\beta_i}} := \arg\min_\beta \frac{1}{N} \|\boldsymbol{\beta_i}^T \boldsymbol{X} - \boldsymbol{z}_i\|_2^2 \tag{19}$$

The point estimation of $\boldsymbol{\beta_i}$ as:

$$\hat{\boldsymbol{\beta}}_i = \left(\boldsymbol{X}^T \boldsymbol{X}\right)^{-1} \boldsymbol{X}^T \boldsymbol{z}_i \tag{20}$$

Without loss of generality, we assume $e_i \sim \mathcal{N}(0, \sigma^2), \forall i$ to derive a loosen bound. In this case, it can be verified that this estimator is unbiased, i.e., $\mathbb{E}[\hat{\boldsymbol{\beta}}_i] = \boldsymbol{\beta}_i$, and the covariance matrix for $\hat{\boldsymbol{\beta}}_i$ is:

$$cov(\hat{\boldsymbol{\beta}}_i) = (\boldsymbol{X}^T \boldsymbol{X})^{-1} \boldsymbol{X}^T cov(z_i) \boldsymbol{X} (\boldsymbol{X}^T \boldsymbol{X})^{-1} = \sigma^2 (\boldsymbol{X}^T \boldsymbol{X})^{-1} \tag{21}$$

Then the mean of $\psi_i(s, a)\hat{z}_i = \psi_i(s, a)\hat{\boldsymbol{\beta}}_i^T \boldsymbol{x}$ is:

$$\mathbb{E}\, \psi_i(s, a)\hat{z}_i = \psi_i(s, a)\hat{z}_i^* \tag{22}$$

and the variance of $\psi_i(s, a)\hat{z}_i = \psi_i(s, a)\hat{\boldsymbol{\beta}}_i^T \boldsymbol{x}$ is:

$$\mathrm{Var}(\psi_i(s, a)\hat{z}_i) = \psi_i^2(s, a)\boldsymbol{x}^T \mathrm{Var}(\hat{\boldsymbol{\beta}}_i)\boldsymbol{x} = \sigma^2 \psi_i^2(s, a)\boldsymbol{x}^T (\boldsymbol{X}^T \boldsymbol{X})^{-1} \boldsymbol{x} \tag{23}$$

Then $\psi_i(s, a)\hat{z}_i$ follows the Gaussian distribution:

$$\psi_i(s, a)\hat{z}_i \sim \mathcal{N}(\psi_i(s, a)\hat{z}_i^*, \sigma^2 \psi_i^2(s, a)\boldsymbol{x}^T (\boldsymbol{X}^T \boldsymbol{X})^{-1}\boldsymbol{x}) \tag{24}$$

Assuming $\hat{z}_i$ are independent of each other, we can get the estimation error of the versatile Q function as:

$$w := \hat{Q}(s, a|\epsilon) - Q^*(s, a|\epsilon) \sim \mathcal{N}\left(0, \sum_{i=1}^M \sigma^2 \psi_i^2(s, a)\boldsymbol{x}^T (\boldsymbol{X}^T \boldsymbol{X})^{-1}\boldsymbol{x}\right) \tag{25}$$

With (25), we can get the prediction error bound as:

$$\Pr\left(-z_{\alpha/2} \leq \frac{\hat{Q}(s, a|\epsilon) - Q^*(s, a|\epsilon)}{\sqrt{\sum_{i=1}^M \sigma^2 \psi_i^2(s, a)\boldsymbol{x}^T (\boldsymbol{X}^T \boldsymbol{X})^{-1}\boldsymbol{x}}} \leq z_{\alpha/2}\right) = 1 - \alpha \tag{26}$$

Also since $\|\psi_{\mathbf{f}, i}(s, a)\| \leq K_{\mathbf{f}}$, with confidence level $1 - \alpha$, we can get:

$$\|\hat{Q}(s, a|\epsilon) - Q^*(s, a|\epsilon)\| \leq z_{\alpha/2}\sqrt{\sigma^2\, M K^2 \boldsymbol{x}^T (\boldsymbol{X}^T \boldsymbol{X})^{-1}\boldsymbol{x}}, \tag{27}$$

where $z_{\alpha/2}$ is the Z-score for the standard Gaussian distributions. Bound shown in (27) depends on the choice of behavior policy conditions (encoded in $\boldsymbol{X}^T \boldsymbol{X}$) and the targeted threshold for adaptation (encoded in $\boldsymbol{x}$).

**(2) Upper bound for all the threshold conditions:** In the next, we are going to find the upper bound for $\sqrt{\boldsymbol{x}^T (\boldsymbol{X}^T \boldsymbol{X})^{-1}\boldsymbol{x}}$ to derive an upper bound for any threshold $\epsilon$. First, we assume the threshold conditions for behavior policies are selected to divide the target condition interval $[\epsilon_L, \epsilon_H]$ evenly. We then show that given limited $N \leq N_{max}$, there exists $B_0(p)$ and $\beta_0(p)$ such that the estimation error bound can be presented as:

$$\|\hat{Q}(s, a|\epsilon) - Q^*(s, a|\epsilon)\| \leq \frac{B_0(p)}{N^{\beta_0(p)}} \tag{28}$$

This result naturally holds since $\psi(s,a)$ is bounded, and $z^*$ can be represented as a polynomial of normalized $\epsilon \in [0,1]$. One trivial solution is a large enough $B(p)$ and $\beta(p) = 0$. The problem is how can we find the tight estimate of $B(p)$ and $\beta(p)$. Here we use the numerical method to find the tight mappings from $p$ to $C$ and $\beta$. When $N \leq 20$, the results are shown as:

$$\sqrt{\boldsymbol{x}^T(\boldsymbol{X}^T\boldsymbol{X})^{-1}\boldsymbol{x}} \leq \frac{B(p)}{N^{\beta(p)}}, \tag{29}$$

where $C(p)$ and $\beta(p)$ can be found in Table. 1. Putting the results in **(1)** and **(2)** together and adding

Table 1: $B(p)$ and $\beta(p)$

| $p$ | 1 | 2 | 3 | 4 | 5 | 6 | 7 | 8 |
|---|---|---|---|---|---|---|---|---|
| $\beta(p)$ | 0.08 | 0.08 | 0.18 | 0.29 | 0.45 | 0.75 | 1.21 | 1.82 |
| $B(p)$ | 1.06 | 1.09 | 1.39 | 2.04 | 3.57 | 9.13 | 37.38 | 248.7 |

the subscript $\mathbf{f}$ for Q functions and the parameter $K_{\mathbf{f}}$, one can derive:

$$\|\hat{Q}_{\mathbf{f}}(s,a|\epsilon) - Q_{\mathbf{f}}^*(s,a|\epsilon)\| \leq \frac{B(p)z_{\alpha/2}}{N^{\beta(p)}}\sqrt{\sigma^2\,MK_{\mathbf{f}}^2}, \tag{30}$$

which is claimed in (17). $\qquad\square$

## C.2  Proof of Proposition 1

*Proof.* With the definition of the versatile safe RL, the optimal policy $\pi^*$ should satisfy the constraint:

$$V_c^{\pi^*(\mu_0|\epsilon)} \leq \epsilon, \tag{31}$$

Consequently, with confidence $1 - \alpha$:

$$\begin{aligned}
V_c^{\pi(\mu_0|\epsilon)} - \epsilon &\leq V_c^{\pi(\mu_0|\epsilon)} - V_c^{\pi^*(\mu_0|\epsilon)} \\
&= \mathbb{E}_{s_0\sim\mu_0,a\sim\pi(\cdot|\epsilon)}\left[Q_c(s,a|\epsilon) - Q_c^*(s,a|\epsilon)\right] \\
&\leq \max_{(s,a)} Q_c(s,a|\epsilon) - Q_c^*(s,a|\epsilon) \\
&\leq \frac{z_{\alpha/2}B(p)}{N^{\beta(p)}}\sqrt{\sigma^2 K_{\mathbf{f}}^2 M},
\end{aligned} \tag{32}$$

where the last inequality comes from Theorem 1. $\qquad\square$

## C.3  Closed-form solution (9)

The closed-form solution is mainly derived from the EM-style safe RL training framework, and more details could be found in Appendix A.2 in [14]. For self-contained, we briefly introduce them as follows. The closed form for E-step optimal variational distribution is:

$$q_i^* = \frac{\pi_{\theta_j}(\cdot|\epsilon_i)}{Z(s,\epsilon_i)}\exp\left(\frac{\hat{Q}_r^{\pi_{\theta_j}}(\cdot|\epsilon_i) - \lambda_i^*\hat{Q}_c^{\pi_{\theta_j}}(\cdot|\epsilon_i)}{\eta_i^*}\right), \tag{33}$$

where $Z(s,\epsilon_i)$ is a normalizer to make sure $q_i^*$ is a valid distribution, and the dual variables $\eta_i^*$ and $\lambda_i^*$ are the solutions of the following convex optimization problem (see Appendix C for details):

$$\min_{\lambda_i,\eta_i\geq 0} g(\eta_i,\lambda_i) = \lambda_i\epsilon_i + \eta_i\kappa\mathbb{E}_{\rho_q}\left[\log\mathbb{E}_{\pi(\cdot|\epsilon_i)}\left[\exp\left(\frac{\hat{Q}_r(\cdot|\epsilon_i) - \lambda_i\hat{Q}_c(\cdot|\epsilon_i)}{\eta_i}\right)\right]\right]. \tag{34}$$

*Proof.* It should be noticed that we have an inherent constraint for $q(\cdot|s,\epsilon_i)$ to be a valid distribution:

$$\int q(a|s,\epsilon_i)da = 1, \quad \forall s \sim \rho_q \tag{35}$$

Then to solve the constrained optimization problem, we first convert it to the equivalent Lagrangian function:

$$L(q, \lambda, \eta, \kappa) = \int \rho_q(s) \int q(a|s) \hat{Q}_r(s, a|\epsilon_i) da ds \tag{36}$$

$$+ \lambda \left( \epsilon_i - \int \rho_q(s) \int q(a|s, \epsilon_i) \hat{Q}_c(s, a|\epsilon_i) da ds \right) \tag{37}$$

$$+ \eta \left( \kappa - \int \rho_q(s) \int q(a|s, \epsilon_i) \log \frac{q(a|s, \epsilon_i)}{\pi(a|s, \epsilon_i)} da ds \right) \tag{38}$$

$$+ \kappa \left( 1 - \int \rho_q(s) \int q(a|s, \epsilon_i) da ds \right), \tag{39}$$

where $\lambda, \eta, \kappa$ are the Lagrange multipliers for the constraints. Since the objective is linear and all constraints are convex (note that KL is convex), the E-step optimization problem is convex. Then we obtain the equivalent dual problem:

$$\min_{\lambda, \eta, \kappa} \max_{q(\cdot|\epsilon_i)} L(q(\cdot|\epsilon_i), \lambda, \eta, \kappa). \tag{40}$$

Take the derivative of the Lagrangian function w.r.t $q$:

$$\frac{\partial L}{\partial q(\cdot|\epsilon_i)} = \hat{Q}_r(s, a|\epsilon_i) - \lambda \hat{Q}_c(s, a|\epsilon_i) - \eta - \kappa - \eta \log \frac{q(a|s, \epsilon_i)}{\pi(a|s, \epsilon_i)}. \tag{41}$$

Let (41) be zero, then we have the form of the optimal $q$ distribution:

$$q^*(a|s, \epsilon_i) = \pi(a|s, \epsilon_i) \exp \left( \frac{\hat{Q}_r(s, a|\epsilon_i) - \lambda \hat{Q}_c(s, a|\epsilon_i)}{\eta} \right) \exp \left( -\frac{\eta + \kappa}{\eta} \right), \tag{42}$$

where $\exp \left( -\frac{\eta + \kappa}{\eta} \right)$ could be viewed as a normalizer for $q(a|s, \epsilon_i)$ since it is a constant that is independent of $q(\cdot|\epsilon_i)$. Thus, we obtain the following form of the normalizer by integrating the optimal $q(\cdot|\epsilon_i)$:

$$\exp \left( \frac{\eta + \kappa}{\eta} \right) = \int \pi_{\theta_i}(a|s) \exp \left( \frac{Q_r^{\pi_{\theta_i}}(s, a) - \lambda Q_c^{\pi_{\theta_i}}(s, a)}{\eta} \right) da, \tag{43}$$

$$\frac{\eta + \kappa}{\eta} = \log \int \pi(a|s, \epsilon_i) \exp \left( \frac{\hat{Q}_r(s, a|\epsilon_i) - \lambda \hat{Q}_c(s, a|\epsilon_i)}{\eta} \right) da. \tag{44}$$

Take the optimal $q$ distribution in Equation (42) and $\frac{\eta + \kappa}{\eta}$ in Equation (44) back to the Lagrangian function (39), we can find that most of the terms are cancelled out, and obtain the dual function $g(\eta, \lambda)$,

$$g(\eta, \lambda) = \lambda \epsilon_i + \eta \kappa + \eta \int \rho_q(s) \log \int \pi(a|s, \epsilon_i) \exp \left( \frac{\hat{Q}_r(s, a|\epsilon_i) - \lambda \hat{Q}_c(s, a|\epsilon_i)}{\eta} \right) da ds. \tag{45}$$

The optimal dual variables are calculated by

$$\eta^*, \lambda^* = \arg \min_{\eta, \lambda} g(\eta, \lambda). \tag{46}$$

$\square$

A good property is that the dual function (46) is convex (as shown in appendix A.3 of [14]), so we could use off-the-shelf convex optimization tools to solve the dual problem.

# D  Supplementary experiments

## D.1  Experiment details

Due to the page limit, we omit some descriptions of experiments in the main context. Here we give the full details of our experiment settings.

**Task.** The simulation environments are from a publicly available benchmark [18]. We consider two tasks (Run and Circle) and four robots (Ball, Car, Drone, and Ant) which have been used in many previous works as the testing ground [10, 11, 12]. For the Circle task, the agents are rewarded for running in a circle but are constrained within a safe region smaller than the target circle's radius. We name the tasks as `Ball-Circle`, `Car-Circle`, `Drone-Circle`, `Drone-Run`, and `Ant-Run`. In the Run tasks, agents are rewarded for running fast between two safety boundaries and are given costs for violation constraints if they run across the boundaries or exceed an agent-specific velocity threshold. The reward and cost functions are defined as:

$$r(\boldsymbol{s_t}) = ||\boldsymbol{x_{t-1}} - \boldsymbol{g}||_2 - ||\boldsymbol{x_t} - \boldsymbol{g}||_2 + r_{robot}(s_t)$$
$$c(\boldsymbol{s_t}) = \mathbf{1}(|y| > y_{lim}) + \mathbf{1}(||\boldsymbol{v_t}||_2 > v_{lim})$$

where $v_{lim}$ is the speed limit, $y_{lim}$ specifies the safety region, $\boldsymbol{v_t} = [v_x, v_y]$ is the velocity of the agent at timestamp $t$, $\boldsymbol{g} = [g_x, g_y]$ is the position of a fictitious target, $\boldsymbol{x_t} = [x_t, y_t]$ is the position of the agent at timestamp $t$, and $r_{robot}(\boldsymbol{s_t})$ is the specific reward for different robot. For example, an ant robot will gain reward if its feet do not collide with each other. In the Circle tasks, the agents are rewarded for running in a circle in a clockwise direction but are constrained to stay within a safe region that is smaller than the radius of the target circle. The reward and cost functions are defined as:

$$r(\boldsymbol{s_t}) = \frac{-y_t v_x + x_t v_y}{1 + |||\boldsymbol{x_t}||_2 - r|} + r_{robot}(\boldsymbol{s_t})$$
$$c(\boldsymbol{s_t}) = \mathbf{1}(|x| > x_{lim})$$

where $r$ is the radius of the circle, and $x_{lim}$ specifies the range of the safety region.

**Constraint-conditioned Baselines.** We design these baselines by directly integrating the threshold as a part of the state in the CMDP tuple, $\bar{s} = [s; \epsilon]$. The policy is optimized with behavior policy conditions only. We adopt commonly used off-policy safe RL algorithms, `SAC-Lag` and `DDPG-Lag`, and name the proposed baselines as `V-SAC-Lag` and `V-DDPG-Lag`.

**Policy linear combination baselines.** We also compare our method with single-threshold policy combinations. Denote $\epsilon$ as an unseen target threshold for adaptation, and $\epsilon_1, \epsilon_2$ as two behavior policy conditions closest to $\epsilon$. Then the policy for $\epsilon$ is the combination of $\pi(\cdot|\epsilon_1), \pi(\cdot|\epsilon_2)$:

$$\pi(\cdot|\epsilon) = w_1\pi(\cdot|\epsilon_1) + w_2\pi(\cdot|\epsilon_2); \quad w_1 = (\epsilon_2 - \epsilon)/(\epsilon_2 - \epsilon_1), \ w_2 = (\epsilon - \epsilon_1)/(\epsilon_2 - \epsilon_1) \quad (47)$$

This method is designed for both threshold interpolation and extrapolation, i.e., the coefficients $w_1$ or $w_2$ can be negative. This baseline draws inspiration from the safe control theory, which suggests the safe input component is proportional to the conservativeness level [27]. To this end, we use two strong on-policy methods `PPO-Lag` and `TRPO-Lag` to train the single-threshold agents and named the corresponding baselines as `C-PPO-Lag` and `C-TRPO-Lag`.

**Metrics:** We compare the methods in terms of episodic reward (the higher, the better) and episodic constraint violation cost (the lower, the better) on each evaluated threshold condition. We take the average of the episodic reward (Avg. R) and constraint violation (Avg. CV) as the main comparison metrics. The constraint violation for threshold $\epsilon$ is defined as: $\text{CV} = \max\{0, \Sigma_\tau c_t - \epsilon\}$. We also report the average performance solely on unseen thresholds (Avg. R-G and Avg. CV-G) to characterize the adaptation capability.

## D.2  Full result table of the main experiment

Due to the page limit, we omit the quantitative results for the experiment. Here we provide the full version of the results and a more details analysis. The full results are shown in Table 2. We utilize the Metrics mentioned in Appendix D.1 for evaluation.

Table 2: Evaluation results of proposed CCPO method and the proposed versatile safe RL baselines. ↑: the higher reward, the better. ↓: the lower constraint violation (minimal 0), the better. The models are evaluated on a series of threshold conditions and we report the averaged reward and constraint violation values on all evaluation thresholds and generalized thresholds. Each value is reported as mean ± standard deviation for 50 episodes and 5 seeds. We shade the two safest agents with the lowest averaged cost violation values.

| Task | Stats | CCPO (ours) | Constraint-conditioned | | Linear combination | |
|---|---|---|---|---|---|---|
| | | | V-SAC | V-DDPG | C-PPO | C-TRPO |
| Ball-Circle | Avg. R ↑ | 710.86±20.47 | 774.16±20.34 | 762.61±58.65 | 637.85±14.03 | 699.38±1.94 |
| | Avg. CV ↓ | 0.59±0.31 | 5.32±5.00 | 2.81±1.12 | 3.11±1.64 | 4.50±0.08 |
| | Avg. R-G ↑ | 699.04±20.48 | 766.52±22.59 | 756.67±58.48 | 667.89±12.17 | 699.14±2.05 |
| | Avg. CV-G ↓ | 0.83±0.42 | 6.29±5.72 | 3.53±1.26 | 3.40±1.75 | 5.59±0.25 |
| Car-Circle | Avg. R ↑ | 406.06±6.30 | 331.80±11.57 | 448.82±18.65 | 440.01±2.59 | 461.14±1.39 |
| | Avg. CV ↓ | 1.60±0.91 | 12.18±4.65 | 14.48±8.14 | 9.09±1.52 | 7.84±1.71 |
| | Avg. R-G ↑ | 401.53±5.59 | 331.19±11.00 | 445.32±17.42 | 438.31±3.03 | 460.72±1.15 |
| | Avg. CV-G ↓ | 1.49±0.38 | 12.74±4.32 | 14.63±8.69 | 11.07±1.58 | 9.14±2.01 |
| Drone-Circle | Avg. R ↑ | 630.55±40.03 | 693.69±22.37 | 734.58±49.69 | 392.64±23.13 | 380.77±18.62 |
| | Avg. CV ↓ | 0.32±0.38 | 13.24±8.80 | 19.62±11.15 | 0.45±0.38 | 6.55±1.95 |
| | Avg. R-G ↑ | 625.51±40.12 | 699.14±24.88 | 730.29±48.43 | 342.77±19.06 | 291.87±19.88 |
| | Avg. CV-G ↓ | 0.47±0.55 | 14.97±10.10 | 19.44±10.36 | 0.21±0.09 | 7.23±2.03 |
| Drone-Run | Avg. R ↑ | 458.69±12.98 | 355.61±35.44 | 244.60±48.29 | 398.88±21.53 | 461.70±4.91 |
| | Avg. CV ↓ | 0.23±0.25 | 8.66±4.30 | 11.33±9.63 | 9.46±5.63 | 47.97±3.49 |
| | Avg. R-G ↑ | 455.64±11.83 | 354.61±33.34 | 236.61±43.49 | 386.77±30.09 | 464.07±6.61 |
| | Avg. CV-G ↓ | 0.33±0.37 | 9.96±4.54 | 12.72±9.91 | 11.18±7.46 | 60.39±4.32 |
| Ant-Run | Avg. R ↑ | 660.88±4.82 | 615.73±91.99 | 594.75±172.35 | 636.06±6.78 | 629.83±7.84 |
| | Avg. CV ↓ | 3.13±1.67 | 8.47±3.55 | 23.69±30.42 | 5.16±1.59 | 0.22±0.17 |
| | Avg. R-G ↑ | 660.07±5.26 | 626.27±84.61 | 592.50±173.01 | 620.46±9.99 | 605.07±10.63 |
| | Avg. CV-G ↓ | 3.25±1.48 | 7.76±11.83 | 22.90±9.39 | 6.73±2.32 | 0.03±0.06 |

## D.3  More experiments with different choices of behavior policy conditions

Due to the page limit, we only provide the experiment results based on behavior policy set $\mathcal{E} = \{20, 40, 60\}$. Here we provide more experiment results when the behavior policy set is selected to be $\hat{\mathcal{E}} = \{10, 30, 50, 70\}$ as shown in Table 3. The algorithms are evaluated on threshold conditions $\mathcal{E} = \{10, 15, ..., 70\}$ for the Ball-Circle, Car-Circle, and Drone-Circle tasks. From the results, we can clearly see that all the conclusions in section 4 also hold for different behavior policy choices.

Table 3: Evaluation results of proposed CCPO method and the proposed versatile safe RL baselines. ↑: the higher reward, the better. ↓: the lower constraint violation (minimal 0), the better. The models are evaluated on a series of threshold conditions and we report the averaged reward and constraint violation values on all evaluation thresholds and generalized thresholds. Each value is reported as mean ± standard deviation for 50 episodes and 5 seeds. We shade the two safest agents with the lowest averaged cost violation values.

| Task | Stats | CCPO (ours) | Constraint-conditioned | | Linear combination | |
|---|---|---|---|---|---|---|
| | | | V-SAC | V-DDPG | C-PPO | C-TRPO |
| Ball-Circle | Avg. R ↑ | 639.65±37.91 | 778.14±7.92 | 737.79±29.84 | 590.58±20.76 | 700.86±2.81 |
| | Avg. CV | 0±0 | 5.98±2.29 | 2.36±1.30 | 0.84±0.58 | 1.78±0.36 |
| | Avg. R-G ↑ | 640.71±37.46 | 781.82±7.73 | 739.22±28.98 | 589.08±18.85 | 702.74±2.93 |
| | Avg. CV-G | 0±0 | 5.95±2.06 | 2.64±1.45 | 0.56±0.48 | 2.28±0.40 |
| Car-Circle | Avg. R ↑ | 414.08±3.47 | 342.68±12.59 | 436.02±33.56 | 440.31±9.83 | 457.25±1.29 |
| | Avg. CV ↓ | 1.18±0.36 | 14.18±6.24 | 19.60±13.67 | 9.49±1.47 | 8.63±1.38 |
| | Avg. R-G ↑ | 414.36±3.17 | 344.02±13.38 | 436.72±33.92 | 441.92±9.16 | 456.23±2.16 |
| | Avg. CV-G ↓ | 1.19±0.35 | 15.23±7.27 | 21.06±14.91 | 11.05±1.82 | 11.24±1.02 |
| Drone-Circle | Avg. R ↑ | 703.06±31.82 | 696.92±35.98 | 719.54±95.85 | 367.84±15.85 | 489.42±9.40 |
| | Avg. CV ↓ | 0±0 | 5.39±4.45 | 11.06±10.12 | 2.09±1.62 | 8.64±1.42 |
| | Avg. R-G ↑ | 705.52±30.35 | 702.72±36.73 | 721.57±95.17 | 277.42±19.26 | 427.59±10.26 |
| | Avg. CV-G ↓ | 0±0 | 5.92±5.03 | 11.37±10.78 | 2.82±1.94 | 12.05±1.74 |

## D.4  $\epsilon$-sampling efficiency evaluation

The proposed algorithm is sampling-efficient as it satisfies the requirement **Thresholds Sampling Efficiency**: it is able to train the versatile safe RL agent with limited behavior policies for data collection. Here we provide some quantitative results. In this experiment, we aim to answer the question: How data-efficient CCPO is compared to exhaustively training the safe RL agents? We compare our method with policy linear combination with different behavior policy set $\tilde{\mathcal{E}} = \{20, 40, 60\}$ and $\tilde{\mathcal{E}}' = \{20, 30, 40, 50, 60, 70\}$, where $|\tilde{\mathcal{E}}'| = 2|\tilde{\mathcal{E}}|$. The algorithms are evaluated on threshold conditions $\mathcal{E} = \{10, 15, ..., 70\}$, and the averaged performance is reported in Table. 4. The safest agent is shadowed. We can clearly see that even if the policy linear combination algorithm C-TRPO uses more behavior policy, it can not behave as safely as the CCPO method on all the evaluated tasks. Thus we can conclude that our method is at least 2 times more $\epsilon$-sampling efficient than exhaustively training the safe RL agents using TRPO-Lag.

Table 4: $\epsilon$-sampling efficiency evaluation. $\uparrow$: the higher reward, the better. $\downarrow$: the lower constraint violation (minimal 0), the better. The models are evaluated on a series of threshold conditions and we report the averaged reward and constraint violation values on all evaluation thresholds and generalized thresholds. Each value is reported as mean ± standard deviation for 50 episodes and 5 seeds. We shade the safest agent with the lowest averaged cost violation value.

| Algorithm | Stats | Ball-Circle | Car-Circle | Drone-Circle | Drone-Run |
|---|---|---|---|---|---|
| CCPO with $\tilde{\mathcal{E}}$ | Avg. R $\uparrow$ | 710.86±20.47 | 406.06±6.30 | 630.55±40.03 | 458.69±12.98 |
| | Avg. CV $\downarrow$ | 0.59±0.31 | 1.60±0.91 | 0.32±0.38 | 0.23±0.25 |
| C-TRPO with $\tilde{\mathcal{E}}$ | Avg. R $\uparrow$ | 699.38±1.94 | 461.14±1.39 | 380.77±18.62 | 461.70±4.91 |
| | Avg. CV $\downarrow$ | 4.50±0.08 | 7.84±1.71 | 6.55±1.95 | 47.97±3.49 |
| C-TRPO with $\tilde{\mathcal{E}}'$ | Avg. R $\uparrow$ | 682.94±8.08 | 458.13±2.22 | 411.91±8.95 | 472.89±2.65 |
| | Avg. CV $\downarrow$ | 2.66±0.37 | 11.90±2.12 | 5.20±0.81 | 30.20±2.47 |

### D.5 Q functions estimation verification

The verification results for the Q function estimation are shown in Figure. 6. The testing task is `Car-Circle`, the behavior policy condition set is $\tilde{\mathcal{E}} = \{20, 40, 60\}$, and the evaluating thresholds are set to be $\{10, 20, 30, 40, 50, 60, 70\}$. We select $Q_c$ as the testing target. The "ground truth" of the Q functions are the Q functions trained with the single-threshold safe RL agents (CVPO). The state-action pair data $(s, a)$ for evaluation are sampled randomly from the replay buffer of one single-threshold CVPO agent. We can see from Figure. 6 that the $Q_c$ distribution mismatch is small on generalized thresholds, which shows that the proposed VVE is efficient in Q function zero-shot adaptation.

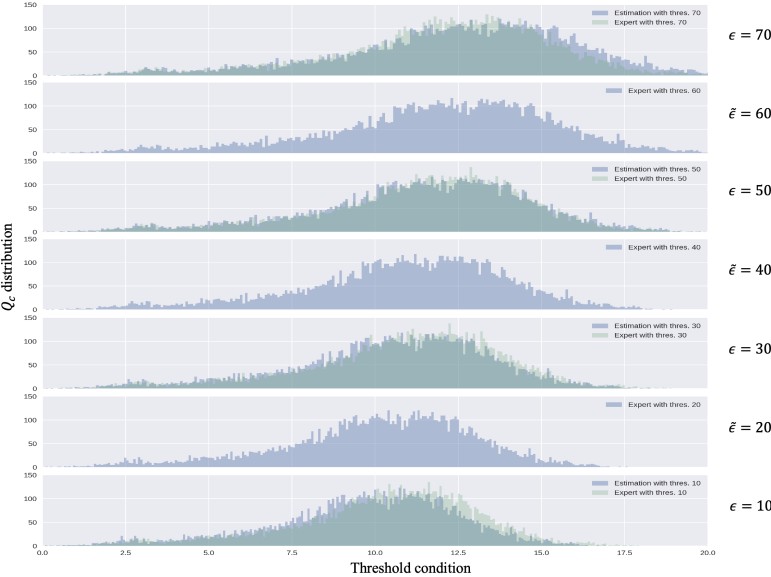

Figure 6: Q function estimation verification. Green histograms represent the evaluation results of the single-threshold policies ("ground truth"), and the blue histograms represent the evaluation results for the versatile Q function.

# E    Related work

**Safe RL** has been approached through various methods. Some techniques leverage domain knowledge of the target problem to enhance the safety of an RL agent [28, 29, 30, 31, 32, 33, 34, 35, 36]. Another line of work employs constrained optimization techniques to learn a constraint-satisfaction policy [37, 1, 38], such as the Lagrangian-based approach [39, 40, 41], where the Lagrange multipliers can be optimized via gradient descent along with the policy parameters [42, 43, 9]. Alternatively, other works approximate the non-convex constrained optimization problem with low-order Taylor expansions [12] or through variational inference [14], then solve for the dual variable using convex optimization [44, 45, 46, 47]. However, most existing approaches consider a fixed constraint threshold during training, which can hardly be deployed for different threshold conditions after training.

**Transfer learning in RL.** The concept of transfer learning, also recognized as knowledge transfer, denotes a sophisticated technique that exploits external knowledge harnessed from various domains to enhance the learning trajectory of a specified target task [48]. Transfer learning in RL can be categorized from multiple perspectives, such as skill composition for novel tasks [49, 50, 51, 52], parameter transfer [53], and feature representation transfer [54, 55, 56]. Among these, the methodologies leveraging Successor Features (SFs) [57, 19] are particularly relevant to this work. These methodologies operate under the assumption that the reward function can be deconstructed into a linear combination of features, and they further extend the successor representation to decouple environmental dynamics from rewards [58, 59]. However, most existing works using SFs consider transfer learning problems among tasks with different reward functions but not with different task conditions.

