# OpenReview forum: "CCPO: Constraint-Conditioned Policy Optimization for Versatile Safe Reinforcement Learning"
_robot-learning.org/CoRL/2023/Workshop/OOD — OOD Workshop @ CoRL 2023_

### Official Review · Reviewer_mM4A · 2023-10-16
**Alignment with OOD generalization seems a bit tenuous**

**Rating:** 6
**Confidence:** 3

**Review:**

This paper provides a constrained RL method that generalizes to new constraint thresholds. The writing and the results of the paper are strong. My only concern is that the connection with OOD generalization seems weak. The training data includes various thresholds in it, so the zero shot adaptation to new thresholds seems to be limited to thresholds sampled from the underlying training distribution from which the training data was drawn, i.e., generalizes only to previously unseen thresholds drawn from the underlying training distribution and not necessarily OOD.

---

### Official Review · Reviewer_rSRB · 2023-10-16
**Weak reject**

**Rating:** 5
**Confidence:** 3

**Review:**

This paper adapts reward maximizing agents to unseen safety requirements allowing zero-shot policy generalization. The authors demonstrate promising initial results showing CCPO achieves comparable or better reward and lower cost relative to baseline algorithms, especially in tasks characterized by highly non-linear dynamics. The paper is well written and sufficiently justifies the novelty of the approach relative to current literature in the field. The proposed problem has broad and relevant application to autonomous agents from autonomous driving to UAV locomotion. A strength of this approach is its flexibility towards arbitrary threshold conditions while a weakness is the indirect application to OOD research through uncertain safety requirements.

---

### Decision · Program_Chairs · 2023-10-17

**Decision:**

Accept

**Comment:**

We agree with the reviewers’ assessment that this work is technically sound and although opinions vary between reviewers, we think this paper will contribute to productive discussions at the 2023 Workshop on OOD Generalization in Robotics. In particular, the reviewers appreciate the strength of the results. However, we would like to stress the reviewers’ comments that the impact of this work (in the context of this workshop) would be improved by highlighting specifically the connection of this work to the core theme of this workshop: Improving reliability in OOD scenarios. We recommend the authors incorporate the reviewers’ feedback into their camera-ready submission to further improve their manuscript.